# Impact of Results of TTF-1 Immunostaining on Efficacy of Platinum-Doublet Chemotherapy in Japanese Patients with Nonsquamous Non-Small-Cell Lung Cancer

**DOI:** 10.3390/jcm12010137

**Published:** 2022-12-24

**Authors:** Akira Nakao, Hiroyuki Inoue, Nobumitsu Ikeuchi, Fumiyasu Igata, Takashi Aoyama, Makoto Hamasaki, Hisatomi Arima, Masaki Fujita

**Affiliations:** 1Department of Respiratory Medicine, Fukuoka University Hospital, Fukuoka 814-0180, Japan; 2Department of Pathology, Fukuoka University School of Medicine, Fukuoka University Hospital, Fukuoka 814-0180, Japan; 3Department of Preventive Medicine & Public Health, Fukuoka University Hospital, Fukuoka 814-0180, Japan

**Keywords:** thyroid transcription factor-1 (TTF-1), pemetrexed, platinum-doublet chemotherapy, biomarker, non-small-cell lung cancer

## Abstract

Background: Pemetrexed is a key drug in chemotherapy for nonsquamous non-small-cell lung cancer (nonsq NSCLC). Several studies have reported thyroid transcription factor-1 (TTF-1) as a biomarker of the efficacy in chemotherapy regimens, including pemetrexed in non-Asian people. Objective: We aimed to examine the impact of the results of the TTF-1 immunostaining of tumor cells on the therapeutic effect of chemotherapy in Japanese patients with nonsq NSCLC. Methods: We examined the results of TTF-1 immunostaining and the clinical background of Japanese patients with nonsq NSCLC who received platinum-doublet chemotherapy at our hospital, from April 2009 to April 2021, and the correlation between regimens with or without pemetrexed in progression-free survival (PFS) and overall survival (OS). The efficacy of each regimen was then compared between TTF-1-positive and TTF-1-negative tumors. Results: TTF-1 immunostaining was performed in 145 patients during the study period: 92 were positive, and 53 were negative. A total of 24 patients presented with EGFR/ALK gene abnormality (16.6%). The PFS and OS of patients who were TTF-1-positive tended to be longer than those of the patients who were TTF-1-negative under either regimen. In other words, patients who were TTF-1-negative were frequently resistant to numerous chemotherapy drugs and experienced a poor prognosis under both regimens. The OS of patients who were TTF-1-positive and treated with the pemetrexed regimen was significantly longer than those on regimens without pemetrexed (963 vs. 412 days, HR = 0.73; 95% CI 0.55–0.96, *p* = 0.022), whereas there was no difference in PFS. Conclusions: The positivity of TTF-1 immunostaining in tumors could be a predominant prognostic marker for patients who have advanced nonsq NSCLC. Our analysis examined the possibility of a pemetrexed regimen leading to a longer prognosis in Asian patients who were TTF-1-positive for nonsq NSCLC, as shown in previous studies.

## 1. Introduction

The recent advances in chemotherapy for lung cancer have been remarkable, especially the emergence of immune checkpoint inhibitors (ICIs), which have greatly changed the standard of care. ICIs are now actively selected as a first-line therapy in the absence of driver gene abnormalities, and although there is hope that ICIs will provide long-term survival, a very different result from that achieved with conventional anticancer agents, it has also been reported that ICIs alone are ineffective in 30–40% of cases [1]. The expression of PD-L1 in tumor cells is used as a biomarker for this effect. Currently, ICIs are used mainly in combination with cytotoxic anticancer agents, except in cases with a high expression of PD-L1. In Japan, many institutions even combine ICIs with cytotoxic agents for patients with high PD expression levels, to reduce disease progression.

The JMDB study reported a subgroup analysis showing that treatment with cisplatin (CDDP) plus pemetrexed (PEM) led to significantly improved results in patients with nonsq NSCLC compared to treatment with CDDP plus gemcitabine, although overall, there was no significant difference in OS in the population with NSCLC [2]. Platinum-PEM therapy, in combination with ICIs, has been used in several large studies in patients with nonsq NSCLC [3,4]. On the other hand, regimens combining platinum agents other than PEM, such as paclitaxel, S-1, and nab-paclitaxel, have similarly shown to be promising options for combination therapy with ICIs [5,6].

The expression of thymidyrate synthase (TS) in tumor tissue, smoking [7], and serum CYFRA levels [8] have been reported as predictive biomarkers for the effect of chemotherapy, including PEM. In addition, TTF-1 immunostaining in tumor tissue may be useful as a predictive factor for the efficacy of regimens combined with PEM [9,10]. Subgroup analyses of large clinical trials (JMDB, PointBreak, PRONOUNCE) using regimens with PEM have reported lower efficacy in TTF-1-negative adenocarcinomas (NOS) [2,11,12]. However, most of these reports are from the US and Europe; reports in Japanese and other Asian populations are limited, and there have been no reports on TTF-1 staining or the prediction of patients’ responses to a regimen with PEM compared with a regimen without PEM therapy.

TTF-1 is involved mainly in the regulation of thyroid-specific genes but also in the activation of lung surfactant and pituitary genes, and it plays an important role in thyroid and lung morphogenesis and in the development and maturation of the central nervous system. The expression of TTF-1 is significantly increased in early and late lung development and is upregulated in lung cell cycle effectors directly related to development [13].

In lung adenocarcinomas, TTF-1 is highly expressed in nonmucinous carcinomas, contributing to increased tumor cell proliferation and survival, and most squamous cell carcinomas of the lung do not express TTF-1, suggesting its utility in the diagnosis and differential diagnosis of lung cancer [14,15,16,17]. The knockdown of TTF-1 inhibits cancer growth [18], and it has also been reported that the manipulation of the TTF-1 promoter to express miR-7 can inhibit the growth of human lung cancer cells [19]. In small-cell lung cancer, conversely, it has been reported that the prognosis is poor in patients with TTF-1 expression, and there is no doubt that TTF-1 has a role as a prognostic factor and can be a therapeutic target [20].

It has also been reported that TTF-1 immunostaining has a prognostic aspect, and TTF-1-negative cases are known to have poor prognoses [21,22,23,24]; this is also shown in the meta-analysis [25]. In light of this, it is possible that TTF-1 is not a predictor of efficacy but rather a prognostic factor. Thus, the verification of this possibility is an important area of research for future studies.

## 2. Patients and Methods

Using medical records, we retrospectively collected information on TTF-1 immunostaining results, patient background, PFS, and OS data for patients with advanced or recurrent nonsq NSCLC who received platinum-doublet chemotherapy as first-line therapy, epidermal growth factor receptor (EGFR), or anaplastic lymphoma kinase (ALK)-tyrosine kinase inhibitor (TKI), at our hospital over a 12-year period, from April 2009 to April 2021. TTF-1 expression was assessed by immunohistochemistry at primary diagnosis as routine diagnostic evaluation (antibody clone 8G7G3/1, Zytomed Systems, dilution 1:200–1:500). Results are reported in a dichotomous manner.

PFS was defined as the time in months from the date of primary diagnosis to the first documented progression of disease, either as radiologically confirmed progression or as death. OS was defined as the time in months from the date of primary diagnosis to death by any cause. Patients who were not in an advanced stage at diagnosis and had undergone radical surgery or chemoradiation were indicated as being in the period after recurrence. The Kaplan–Meier method was used for the analysis of PFS and OS. The Mann–Whitney U test was performed for univariate analyses, while a logistic regression analysis and the Cox proportional hazards model were performed as multivariable analyses for other items. All analyses were performed using EZR. Two-tailed *p* values of < 0.05 were considered statistically significant.

## 3. Results

In total, 275 patients with nonsq NSCLC were treated with platinum-doublet chemotherapy in our department during the study period. Further, 130 cases in which TTF-1 immunostaining of tumor tissue was not performed were excluded. In accordance with the WHO Classification of Tumors of the Lung, Pleura, Thymus, and Heart, at our hospital, if adenocarcinoma or squamous is diagnosed at the time of hematoxylin and eosin (HE) staining of tumor tissue, no further immunostaining, including TTF-1, is performed. In addition, TTF-1 staining is no longer performed in all cases, because more tumor tissues are now used for driver gene testing and PD-L1 immunostaining.

The remaining 145 cases in which TTF-1 immunostaining was performed were included in this study. Of them 92 cases were positive for TTF-1, and 53 cases were negative. Among the patients who were TTF-1-positive, 43 patients received platinum-doublet chemotherapy with pemetrexed (PEM+ group), and 49 patients received platinum-doublet chemotherapy without pemetrexed (non-PEM+ group). Among the patients who were TTF-1-negative, 21 patients received platinum-doublet chemotherapy with pemetrexed (PEM− group), and 32 (non-PEM− group) received platinum-doublet chemotherapy without pemetrexed (Figure 1). The most common chemotherapy regimen in each group was CBDCA+nab-PTX in the non-PEM− group (16 cases; 50.0%) and the non-PEM + group (21 cases; 42.9), CBDCA + PEM or CBDCA + PEM + Bev in the PEM− group (6 cases; 28.6%, respectively), and CBDCA + PEM in the PEM + group (23 cases; 53.5%). A detailed description of the chemotherapy regimen is provided in Appendix A.

The breakdown of histological subtypes other than adenocarcinoma for patients who were TTF-1-negative was as follows: 14 NOS, 2 pleomorphic carcinomas, and 1 LC-NEC in the non-PEM− group; 1 NOS, 3 pleomorphic carcinomas, and 1 LC-NEC + adenocarcinoma mixed type in the non-PEM+ group; 3 NOS and 1 LC-NEC in the PEM− group; 2 NOS, 1 pleomorphic carcinoma; and 1 pleomorphic carcinoma + adenocarcinoma mixed type in the PEM+ group.

### 3.1. Examination of the Relationship between the Clinical Background and TTF-1 Immunostaining

Differences in patient background factors between TTF-1-positive cases and TTF-1-negative cases were evaluated. The following factors were considered: age, sex, smoking history, number of pack-years, histological subtype, clinical stage, presence of EGFR/ALK gene abnormalities, type of platinum agent (cisplatin (CDDP) or carboplatin (CBDCA)), concomitant use of bevacizumab (Bev), concomitant use of immune checkpoint inhibitor (ICI), renal function (serum estimated glomerular filtration rate (eGFR) at the start of treatment), and presence of interstitial lung disease. The univariate analyses revealed differences in the number of pack-years, histology, EGFR/ALK gene abnormalities, and interstitial pneumonia. The multivariable analysis of these variables revealed a lower rate of adenocarcinoma in TTF-1-negative cases in comparison to TTF-1-positive cases (Table 1).

### 3.2. TTF-1 Immunostaining and Effects of Platinum-Doublet Chemotherapy Regimens; TTF1-Positive vs. TTF-1-Negative

The PFS and OS of patients who were TTF-1-positive and patients who were TTF-1-negative treated with platinum-doublet chemotherapy regimens with and without PEM were examined. In the PEM+ and PEM− groups, PFS was 5.6 months and 3.8 months, respectively (HR = 0.55; 95% CI 0.31–0.97, *p* = 0.037), while OS was 32.1 months and 10.5 months (HR = 0.26; 95% CI 0.13–0.52, *p* < 0.001), respectively. For regimens without PEM, the PFS of patients in the non-PEM+ and non-PEM− groups was 5.2 months and 4.2 months, respectively (HR = 0.53; 95% CI 0.33–0.87, *p* = 0.010), while the OS was 13.7 months and 10.2 months (HR = 0.61; 95% CI 0.36–1.05, *p* = 0.072) (Figure 2A,B), respectively. Although this study excluded cases in which TTF-1 immunostaining had not been performed in patients diagnosed with adenocarcinoma in HE staining, the response of these patients to platinum-doublet chemotherapy was very similar to that of patients who were TTF-1-positive (Appendix A).

In addition, a multivariable analysis was performed to eliminate, as much as possible, the influence of other patient characteristics. For PFS and OS, a univariable analysis showed that both PFS and OS were significantly better in patients with the histological subtype of adenocarcinoma and patients who were TTF-1-positive. As confirmed in Table 1, the histological subtype and the results of TTF-1 immunostaining are strongly correlated. We selected clinical stage and the presence or absence of EGFR/ALK gene abnormalities, in addition to TTF-1, as explanatory variables in the multivariable analysis. The multivariable analysis showed significant differences in the OS between the PEM+ and PEM− groups (HR, 0.26; 95% CI, 0.14–0.58; *p* < 0.001). In contrast, the OS of the non-PEM+ and non-PEM− groups did not differ, according to the results of TTF-1 immunostaining (HR 0.60; 95% CI, 0.35–1.04; *p* = 0.068) (Table 2).

### 3.3. TTF-1 Immunostaining and Effects of Platinum-Doublet Chemotherapy Regimens: PEM vs. Non-PEM

Both the PFS and the OS in each subgroup of patients who were TTF-1-positive and TTF-1-negative with nonsq NSCLC were compared for regimens with PEM and those without PEM. PFS was not significantly different between PEM+ and non-PEM+ groups (HR = 1.14; 95% CI 0.91–1.42, *p* = 0.243), but OS was significantly longer (HR = 0.73; 95% CI 0.55–0.96, *p* = 0.022) in the PEM+ group than in the non-PEM+ group. For patients who were TTF-1-negative, there was no difference in PFS or OS between the PEM− and non-PEM− groups (PFS; HR = 1.14; 95% CI 0.65–2.03, *p* = 0.638, OS; HR = 1.05; 95% CI 0.57–1.95, *p* = 0.871) (Figure 2A,B).

The multivariable analysis of each subgroup showed that patients who were TTF-1-positive in the PEM+ group had significantly better OS than those in the non-PEM+ group (HR = 0.74; 95% CI 0.55–0.99, *p* = 0.039) (Table 3).

### 3.4. (Additional Analysis) Treatment Strategies for TTF-1-Negative Nonsq NSCLC

When the current standard regimen of platinum-doublet chemotherapy plus ICIs or bevacizumab was examined in TTF-1-negative cases, as noted above, there was no significant difference. However, when factoring in the drugs administered in combination with platinum-ICIs, there was a trend toward non-PEM agents, i.e., taxane being better than PEM (Appendix A). In addition, combinations with bevacizumab tended to prolong PFS and OS, although not to a statistically significant extent, and this trend was maintained by the combination of platinum-doublet chemotherapy, ICIs, and bevacizumab (Appendix A). These results indicate that it may be worthwhile to further investigate regimens combining platinum and taxane or even bevacizumab for patients who are TTF-1-negative.

## 4. Discussion

There were three main findings of the present study. First, nonsq NSCLC cases that are TTF-1-positive are more likely to be adenocarcinoma in comparison to those that are TTF-1-negative. Second, patients who were TTF-1-positive tended to have longer PFS and OS than patients who were TTF-1-negative in either regimen with or without PEM, indicating that that TTF-1-negative lung cancer may be resistant to chemotherapy. Consequently, as TTF-1 negativity likely acts as a prognostic factor, the decision whether to administer PEM should not be based on the results of TTF-1 immunostaining. Third, patients on regimens with PEM had a significantly longer OS than those on regimens without PEM, and there was a certain survival benefit seen for patients who were TTF-1-positive.

Three reports that classified patients into the same four groups as this study have been published thus far. A study on stage-four lung adenocarcinoma from the United States [26] reported that TTF-1 immunostaining was a prognostic factor rather than a predictor of the response to platinum-doublet chemotherapy including PEM. The other was a study from Germany on EGFR/ALK-gene-aberration-negative non-small-cell lung cancer [9], which reported that TTF-1-positive cases responded better to regimens with PEM and that TTF-1-negative cases responded better to regimens without PEM; thus, TTF-1 status could be a biomarker for antitumor efficacy. Additionally, another subgroup analysis of the PointBreak trial with a similar design was reported, although it was only presented at a conference [27]. Notably, our study is the first report from an Asian country analyzing the impact of TTF-1 status on the clinical efficacy of platinum-doublet chemotherapy with or without PEM.

First, we will discuss the high rate of adenocarcinomas in TTF-1-positive cases. It was reported that this is due to differences in the carcinogenic process between TTF-1-positive and TTF-1-negative tumors. Lung adenocarcinomas are classified into two categories, on the basis of their origin: terminal respiratory unit (TRU) carcinomas and non-TRU carcinomas. TRU carcinomas are TTF-1-positive, highly differentiated tumors that arise from type 2 alveolar epithelial cells and Clara cells, whereas non-TRU carcinomas are TTF-1-negative tumors that arise from dysplastic mucosa columnar cells located in the central part of non-TRU carcinomas, which often coexist with squamous cell carcinomas (sqCC). The examination of surgical specimens suggests that non-TRU carcinomas not only are centrally located, like sqCCs, but also have a common origin in the epithelium of the bronchial surface, which is likewise strongly associated with tobacco use [28]. Thus, TTF-1-negative tumors may behave similarly to sqCC. To confirm whether the results in this study were influenced by including the subpopulation of nonadenocarcinoma, such as NOS and LC-NEC, in both patients who were TTF-1-positive and those who were TTF-1-negative, we performed an additional survival analysis on the histologic subtype of adenocarcinoma alone, but the obtained results were very similar to those presented herein, which leads us to believe that TTF-1 could be a prognostic factor (Appendix A).

Second, we will discuss the role of TTF-1 immunostaining as a prognostic marker of PFS and OS in chemotherapy for nonsq NSCLC, although the number of reports on its association with the response to PEM has increased in recent years. It is hypothesized that TTF-1-negative tumors may be a factor in a poor response to PEM, as several reports have shown that PEM is less effective in smokers and that TTF-1 may reflect the effect of smoking [20], and it is also known that the expression of TTF-1 is strongly correlated with the degree of differentiation in tumor tissue [29] and that disease behavior similar to that of sqCC may be associated with a poorer prognosis in comparison to patients with nonsq NSCLC [30,31]. In this study, no differences were observed in patient background, so the difference may have been attributable to the number of cases. Furthermore, the possibility that racial differences were reflected cannot be denied.

We will discuss the resistance of TTF-1-negative tumors to platinum doublet. In recent years, TTF-1 immunostaining and the efficacy of PEM have been prominently reported; however, there is also a report noting that docetaxel is less effective in patients who are TTF-1-negative [32] and that patients who are TTF-1-negative are less likely to respond to bevacizumab [33] and EGFR-TKI [34]. ICIs have been reported to be less effective in patients who were TTF-1-negative [35]. It has recently been shown the inactivation of the KEAP1 gene was more frequent in TTF-1-negative tumors [36]. The inactivation of KEAP1 promotes cancer growth and metastasis [37] and is associated with resistance to not only cytotoxic anticancer agents and radiotherapy but also ICIs because the inactivation of the KEAP1 gene results in immunologically cold tumors [38].

In addition, we have to consider the fact that there was a significant difference in OS when comparing regimens with PEM and without PEM, even though there was no significant difference in PFS. In other words, regimens with PEM do not prolong the PFS of patients because of the effects of the platinum doublet itself, but rather, pre-treatment or post-treatment regimens may significantly contribute to prolonged OS.

The most important feature of our study is the inclusion of cases in which ICIs were combined with platinum-doublet chemotherapy, which was not included in previous studies, even though the statistics did not show a significant difference between the regimens with and without ICIs. ICIs are antibodies with a long half-life [39], and the possibility of improving the efficacy of salvage chemotherapy with ICI use in patients with NSCLC has been reported [40]. Recent studies reported that PEM treatment can induce more immunogenic cell death by releasing more damage-associated molecular patterns (DAMPs) from tumor cells than other cytotoxic anticancer drugs, as evidenced by the detection of substantial DAMPs in plasma derived from patients receiving PEM as well as in multiple human NSCLC cell lines [41,42], thereby suggesting that ICI therapy in combination with PEM may improve OS.

The median OS of 32.1 months for the patients who were TTF-1-positive who received regimens with PEM observed in the present study is much longer than those found in three previous reports (16.0 months in Schilsky et al. [26], 12.3 months in Frost et al. [9], and 17.6 months in Garon et al. [27]). The OSs in these studies may have been reduced because Frost et al. excluded cases with EGFR and ALK gene abnormalities. On the other hand, in the clinical studies reported by Garon et al. and Schilsky et al., 14.1% and 20.0% of the patients with nonsq NSCLC were EGFR/ALK-mutation-positive, which is similar to the subpopulation rate in the present study (16.6%), so we do not believe that this demonstrates significant selection bias for a clinical benefit. Although, as discussed before [1], the fact that patients who were TTF-1-positive tended to have more driver EGFR and ALK gene abnormalities, as shown in Table 2 and Table 3; a multivariable analysis showed that the presence or absence of EGFR/ALK gene abnormalities had no significant effect on either PFS or OS, and analyses of the four groups excluding EGFR/ALK-positive patients were also performed. The overall trend was unchanged. For reference, the median OS in the PEM+ group was 32.1 months, and there was no change in the median value here either (Appendix A).

PEM is a folate metabolism antagonist and acts by inhibiting thymidylate synthase (TS), dihydrofolate reductase (DHFR), and glycinamide ribonucleotide formyltransferase (GARFT TTF-1). It has been shown to regulate tumor cell growth and metastasis via a variety of downstream target genes, including Selenbp1, EGFR, Foxa2, CDX2, and DDB1 [43,44,45,46]. However, we were not able to find any studies investigating the relationship between these genes and the aforementioned folate-metabolizing enzymes; thus, we believe that further studies are needed.

Finally, the optimal treatment strategy for TTF-1-negative tumors should be considered in the future. The high likelihood that taxane-based anticancer agents will be effective regardless of TTF-1 staining results [9] was not replicated in our study, suggesting that the selection of the chemotherapy regimen should be based on other clinical factors. In this context, the combination of a platinum agent, taxane-based drugs, and ICIs was considered to have significant potential. Thus, studies on the efficacy and safety of regimens selected on the basis of TTF-1 staining results and combination therapy can be expected. Additionally, the concomitant use of the anti-VEGF antibody bevacizumab may also have a positive effect, contrary to previous reports [33]. As for angiogenesis inhibitors, TTF-1 is involved in VEGF-mediated angiogenesis through the Nrf2 pathway, and although the analysis of a small number of cases from Japan showed that the additive effect was limited, there is still room for further investigation.

## 5. Limitation

This study was a single-center, retrospective analysis and included a relatively small number of cases. Additionally, it is necessary to take into account that the main analysis included patients with EGFR/ALK gene abnormalities, who are considered to be more likely to respond to regimens with PEM. Although these patients accounted for a minority of those included, the ALK gene test was not approved in Japan until 2012, and thus patients with undetected cases may have been included.

## 6. Conclusions

Our retrospective study collectively showed that TTF-1 immunostaining of tumor cell seemed to be a prognostic marker rather than a predictive marker. Our results also suggested that Asian people with TTF-1-positive nonsq NSCLC, especially Japanese patients, may benefit from a regimen with PEM, and a combination with ICIs may lead to further prolonged survival. On the other hand, it could be assumed that patients who were TTF-1-negative were resistant to numerous cytotoxic chemotherapeutic agents and may not benefit from the use of PEM when even patients with nonsq NSCLC have resulting interstitial pneumonia or renal dysfunction. Further investigations to develop an optimal treatment strategy including ICIs or antiangiogenic therapy, e.g., bevacizumab, for patients who are TTF-1-negative are warranted in the future.

## Figures and Tables

**Figure 1 jcm-12-00137-f001:**
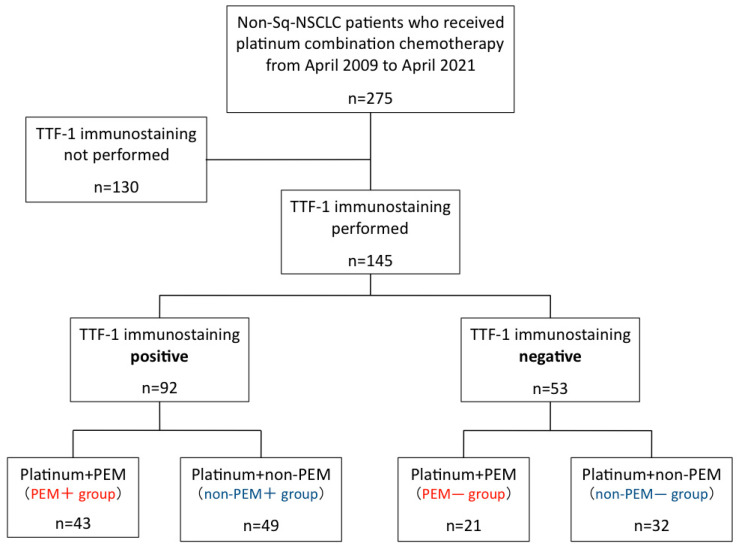
Consort diagram. Abbreviations: sq—squamous cell carcinoma; NSCLC—non-small-cell lung cancer; TTF-1—thyroid transcription factor-1; PEM—pemetrexed.

**Figure 2 jcm-12-00137-f002:**
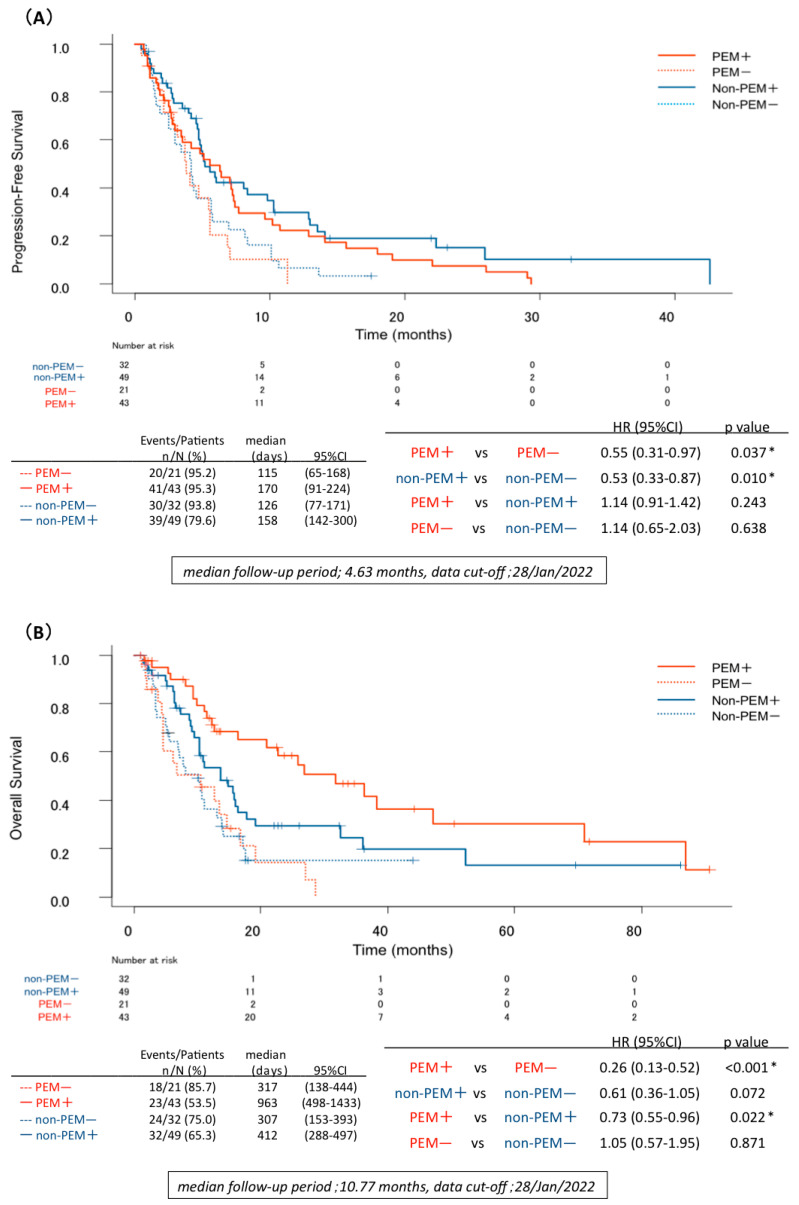
Kaplan–Meier curves of (**A**) PFS and (**B**) OS for each group. Abbreviations: PFS—progression-free survival; OS—overall survival; PEM—pemetrexed; *—statistically significant.

**Table 1 jcm-12-00137-t001:** Patient characteristics.

		Overall	TTF-1 (+)	TTF-1 (−)	*p* Value	*p* Value
		(*n* = 145)	(*n* = 92)	(*n* = 53)	(Univariate)	(Multivariable)
**Age**	Median (Range)	68 (37–84)	68 (37–84)	67 (44–82)	0.799	
**Sex**	Male	118	74	44	0.704	
	Female	27	18	9		
**Smoking history**	Ever	118	72	46	0.206	
	Never	27	20	7		
**pack-year**	Median (Range)	38 (0–175)	35 (0–132.5)	45 (0–175)	0.018 *	0.455
**Histology**	Adeno	115	84	32	<0.0001 *	<0.001 *
	Others	29	8	21		
**Disease stage**	3	14	6	8	0.703	
	4	112	72	40	0.324	
	Recurrence	19	14	5		
**EGFR/ALK gene abnormality**	Present	24	21	3	0.008 *	0.162
	Absent	121	71	50		
**Types of platinum agents**	Cisplatin	21	13	8	0.877	
	Carboplatin	124	79	45		
**Concomitant use of Bev**	Yes	38	27	11	0.26	
	No	107	65	42		
**Concomitant use of ICI**	Yes	39	26	13	0.629	
	No	106	66	40		
**eGFR at** **diagnosis**	Median (Range)	72 (4–164)	72 (4–64)	72 (45–126)	0.757	
**Presence of ILD**	Yes	15	6	9	0.048 *	0.079
	No	130	86	44		

Abbreviations: EGFR—epidermal growth factor receptor; ALK—anaplastic lymphoma kinase; Bev—bevacizumab; ICI—immune checkpoint inhibitor; eGFR—estimated glomerular filtration rate; ILD—interstitial lung disease; *—statistically significant.

**Table 2 jcm-12-00137-t002:** Multivariable analysis of PFS and OS in regimen with and without PEM.

	Platinum-Pemetrexed (*n* = 64)				Platinum-non-Pemetrexed (*n* = 81)			
	PFS HR (95% CI)	PFS *p* Value	OS HR (95% CI)	OS *p* Value	PFS HR (95% CI)	PFS *p* Value	OS HR (95% CI)	OS *p* Value
**EGFR/ALK wild type**	1.00	1.00	1.00	1.00	1.00	1.00	1.00	1.00
**EGFR/ALK mutated**	0.87	0.658	0.75	0.484	1.33	0.462	1.07	0.879
	(0.48–1.60)		(0.34–1.68)		(0.62–2.84)		(0.45–2.55)	
**Stage3/** **recurrence**	1.00	1.00	1.00	1.00	1.00	1.00	1.00	1.00
**Stage4**	0.86	0.634	1.04	0.918	0.56	0.051	0.58	0.086
	(0.47–1.59)		(0.50–2.16)		(0.32–1.00)		(0.32–1.08)	
**TTF-1** **negative**	1.00	1.00	1.00	1.00	1.00	1.00	1.00	1.00
**TTF-1** **positive**	0.75	0.059	0.54	<0.001 *	0.58	0.019 *	0.6	0.068
	(0.56–1.01)		(0.37–0.77)		(0.35–0.96)		(0.35–1.04)	

Abbreviations: EGFR—epidermal growth factor receptor; ALK—anaplastic lymphoma kinase; PFS—progression-free survival; OS—overall survival; HR—hazard ratio; CI—confidence interval; *—statistically significant.

**Table 3 jcm-12-00137-t003:** Multivariable analysis of PFS and OS in patients who were TTF-1-positive and patients who were TTF-1-negative.

	TTF-1 Positive (*n* = 92)				TTF-1 Negative (*n* = 53)			
	PFS HR (95% CI)	PFS *p* Value	OS HR (95% CI)	OS *p* Value	PFS HR (95% CI)	PFS *p* Value	OS HR (95% CI)	OS *p* Value
**EGFR/ALK** **wild type**	1.00	1.00	1.00	1.00	1.00	1.00	1.00	1.00
**EGFR/ALK** **mutated**	1.18	0.554	0.84	0.601	0.52	0.322	0.97	0.964
	(0.68–2.01)		(0.44–1.62)		(0.14–1.89)		(0.22–4.34)	
**Stage3/** **recurrence**	1.00	1.00	1.00	1.00	1.00	1.00	1.00	1.00
**Stage4**	0.65	0.109	0.7	0.27	0.57	0.136	0.74	0.417
	(0.38–1.10)		(0.37–1.32)		(0.28–1.19)		(0.36–1.52)	
**Chemotherapy without PEM**	1.00	1.00	1.00	1.00	1.00	1.00	1.00	1.00
**Chemotherapy with PEM**	1.09	0.486	0.74	0.039 *	1.21	0.516	1.01	0.972
	(0.86–1.38)		(0.55–0.99)		(0.67–2.16)		(0.54–1.90)	

Abbreviations: EGFR—epidermal growth factor receptor; ALK—anaplastic lymphoma kinase; PFS—progression-free survival; OS—overall survival; HR—hazard ratio; CI—confidence interval; *—statistically significant.

## Data Availability

The datasets generated and/or analyzed during the current study are available from the corresponding author on reasonable request.

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
