# Peer review of "Impact of Results of TTF-1 Immunostaining on Efficacy of Platinum-Doublet Chemotherapy in Japanese Patients with Nonsquamous Non-Small-Cell Lung Cancer"

_jcm, 2022, doi:10.3390/jcm12010137_

Round 1

Reviewer 1 Report

The manuscript of “Correlation of Results of TTF-1 Immunostaining with Efficacy 2 of Chemotherapy in Japanese Patients with Non-Squamous 3 Non-Small Cell Lung Cancer” by Drs. Akira Nakao et al reports the results of Thyroid Transcription Factor-1 11 (TTF-1) as a biomarker for use for chemotherapy regimens including pemetrexed in Japanese Patients with Non-Squamous Non-Small Cell Lung Cancer. TTF-1 immunostaining was performed in 145 patients with 92 were positive and 53 were negative. The comparison of TTF-1-positive and TTF-1-negative patients revealed that PFS was significantly shorter in either regimens (170 vs 115days, p=0.037 in pemetrexed-containing regimens) , (158 vs 126days, p=0.010 in regimens with-out pemetrexed), and OS (963 vs 317days, p<0.001) were better in TTF-1-positive patients. They conclude that the possibility of a longer prognosis with a pemetrexed-containing regimen in TTF-1-positive Asian non-Sq NSCLC patients.

Overall, the manuscript was well organized. However. there are some issues for the analysis and explanation of the data.

Major concerns:

1.     Figure 2 A-D show that TTF-1 positive patients have better PFS and OS in both Pem treated and non-pem treated. To show that if differential TTF-1 effect on the outcome, interaction between the TTF-1 status and treatment should be tested. If no significant differential effect, then TTF-1 should be reported as a prognostic factor for the patient’s population, not the limited subgroup. Meanwhile, please report the estimated hazard ratio (HR) and its 95% C.I., not just the median and p-value, as the p-values are corresponding the estimated HRs.

2.     In table 2, the reference group should be provided in the table. Some results are of concern in the table, i.e., the estimated HR of Stage 4 is less than 1 (compare to stage 3), which mean they have better outcome in PFS and OS than those patients with Stage 3 disease, contrary to the common knowledge on the disease.

3.     If the reported results in table 2 are the multivariable model with all those variables in the model, then, the model is over fitted, caused the issue in 2.

4.     Please report the follow up time and the number of events for both PFS and OS.

Minors:

1.     In  table 2, the number of patients (69) is different from the non-pem treated patients (49+32 = 81), is it due to missing values in covariates?

Author Response

Thank you for the helpful comments on our manuscript. Please see the attachment.

Reviewer 2 Report

Nakao and colleagues investigated the impact of TTF1 expression on PFS and OS in a monocentric study of japanese patients. While substantial differences were observed in TTF1+ patients according to the regimens applied, outcome in TTF1- patients appeared to be similar. All in all, the results are interesting, but I have some doubts concerning the meaningfulness of the data, all with regard to substantial selection bias: First, given a long recruitment period (12 years) the number of patients involved is low and half of those had no information on TTF1 staining available. Second, the authors also included "other" histologies (what were these?), partly as a relevant share (all in all ~25%). Third, patients with targetable genomic alterations (ALK, EGFR) were not excluded (this was not statistically significant, probably due to the low number of cases included) which may represent an additional bias given better responses to pemetrexed-based regimens in this specific population (reported e. g. for ALK/ROS/RET+ patients). In this connection, the reported better outcome in PEM+ regimens must at least be questioned. 

The main difficulty of the results is that the principle goal of the study (Correlation of Results of TTF-1 Immunostaining with Efficacy of Chemotherapy) is not really reflected within the results section. Outcome (OS, PFS) is presented according to the regimens applied (PEM+ vs. PEM-) and TTF1 expression is only included as one variable in the Cox proportional hazard regression analysis (among multiple others). The relevant information supporting the hypothesis of different outcome according to the regimens applied can only be found in the supplement (Fig. 1a/b).

The manuscript is hard to read and it was partly difficult to understand the writers' intentions.

Author Response

(The authors gave the same response as above.)

Round 2

Reviewer 1 Report

The manuscript of “Correlation of Results of TTF-1 Immunostaining with Efficacy 2 of Chemotherapy in Japanese Patients with Non-Squamous 3 Non-Small Cell Lung Cancer” by Drs. Akira Nakao et al reports the results of Thyroid Transcription Factor-1 11 (TTF-1) as a biomarker for use for chemotherapy regimens including pemetrexed in Japanese Patients with Non-Squamous Non-Small Cell Lung Cancer.
The revision of the manuscript answers some questions, but some issues remain there. As those are all relevant to statistical analysis, the authors are strongly recommended to consult with a statistician for the analysis. Here are some remained issues.
Major concerns:
1. In line 190, (HR=1.14; 95%CI 0.33--0.87,0.87, p=0.010), the HR= 1.14 is out of the 95% C.I., either the HR of 1.14 or the C.I. is wrong.
2. In line 206, the reported (HR, 0.28; 95%CI, 0.14-0.58; p<0.001) is different from what is given in table 2.
3. In table 3, the analysis was performed in TTF-1 positive and negative separately, where are those comparison between TTF-1+ and TF-1- come from?

Reviewer 2 Report

The authors have substantially improved the manuscript. However, with regard to histologic subtypes I suggest to exclude LCNEC and NOS (major discrepances e. g. between Non-PEM- and the other groups) as this makes the interpretation of the results even more difficult. A clear prognostic value of TTF-1 has only be demonstrated in lung adenocarcinoma (LuAD) and, albeit most of NOS carcinoma might be de-differentiated LuAD, some will originate from squamous carcinoma.
